# Experimental Study of Military Crawl as a Special Type of Human Quadripedal Automatic Locomotion

**Dmitry Skvortsov** [1,2,*] **, Victor Anisimov** [2,3,*] **and Alina Aizenshtein** [2,4]

1. Federal Scientific and Clinical Center for Specialized Medical Assistance and Medical Technologies of the Federal Medical Biological Agency, 115682 Moscow, Russia
2. Dmitry Rogachev National Medical Research Center of Pediatric Hematology, Oncology and Immunology, 117997 Moscow, Russia; kagina19@mail.ru
3. Faculty of Biology, Lomonosov Moscow State University, 119991 Moscow, Russia
4. Federal Center of Brain and Neurotechnology, 117997 Moscow, Russia
* Correspondence: skvortsov.biom@gmail.com (D.S.); victor_anisimov@neurobiology.ru (V.A.); Tel.: +7-916-692-5419 (D.S.); +7-926-519-6801 (V.A.)

**Featured Application: The results of this study can be useful for further research into features related to the development of *military crawl* locomotion in groups of different ages, and for understanding the development of this type of locomotion in ontogenesis. In rehabilitation clinics, it is possible to compare patients' indicators with the norm registered in this study (which could be further expanded), to assess the degree of violations in the motor control system.**

**Abstract:** The biomechanics of military crawl locomotion is poorly covered in scientific literature so far. Crawl locomotion may be used as a testing procedure which allows for the detection of not only obvious, but also hidden locomotor dysfunctions. The aim of the study was to investigate the biomechanics of crawling among healthy adult participants. Eight healthy adults aged 15–31 (four women and four men) were examined by means of a 3D kinematic analysis with Optitrack optical motion-capture system which consists of 12 Flex 13 cameras. The movements of the shoulder, elbow, knee, and hip joints were recorded. A person was asked to crawl 4 m on his/her belly. The obtained results including space-time data let us characterize military crawling in terms of pelvic and lower limb motions as a movement similar to walking but at a more primitive level. Progressive and propulsive motions are characterized as normal; additional right–left side motions—with high degree of reciprocity. It was found that variability of the left-side motions is significantly lower than that of the right side (Z = 4.49, $p < 0.0001$). The given normative data may be used as a standard to estimate the test results for patients with various pathologies of motor control (ataxia, abasia, etc.).

**Keywords:** crawling; biomechanics; kinematics of motion; 3D kinematic analysis

## 1. Introduction

For a human being crawling is a particular stage of phylo- and onto-genesis. This locomotion consists of phylogenetically old and well-automated motions, which human beings acquire at an early stage of their ontogeny and could be described as a type of quadripedal automatic locomotion [1,2]. The locomotion itself is used in a number of physical therapy systems, for instance, in the Klapp system [3]. Nowadays, crawling is mostly applied for a military purpose and also as a fitness exercise. Crawling distinguishes several types: on one's side, belly crawl (military crawl) and on hands and knees (bear crawl). The term crawling itself is used to denote several different locomotion types. Normally it refers to moving on hands, legs and belly, which would be more correct to call military crawling (Figure 1).

A more detailed literature search has shown that the term crawling may be used for other locomotion types as well. In the English literature, there are two terms used

to describe this movement. The common word for this is crawl. Like any word in the English language, this one has plenty of meanings as well. It is used to describe all crawl types and some styles in swimming. Each connotation is interpreted based on the context. Another term is creeping, which is used for the same purpose but may have a limited context. This term is used to describe moving on hands and knees, and sometimes by way of military crawling.

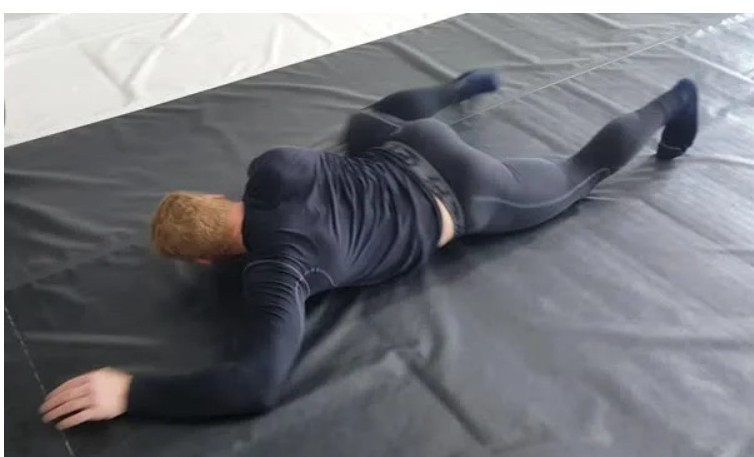

**Figure 1.** Example of military crawling.

In the modern literature, there are the following types of crawling used by babies and toddlers at different developmental stages: moving on hands and knees (diagonal limb functioning), moving on hands and feet, moving on belly, scooting (moving in sitting position with the help of one's hands), which is a mix of the first two types (one hand and foot on one side and one hand and knee on the other side [3,4]), and a mixed way of using three limbs as the main support.

Nevertheless, after analyzing different or slightly similar samples of babies' crawl there comes a conclusion that, despite the high variability of the applied crawl techniques, all of them have certain constant values inside the interlimb coordination [4]. A pace such as trot prevails even if only three limbs are used [4,5]. It may be confined to the synchronized functioning of limbs, which is possible only when the nervous system is mature.

In study [6], the above-mentioned group of healthy participants showed that the trot pace used when one moves on hands and knees turned out to be more constant and frequent, but at a lower speed. At the same time, while the body is gradually moving forward, it experiences minimal fluctuations.

In paper [1], the authors came to a conclusion about the mechanisms of moving on all four limbs using patterns characteristic of both apes and other animals. Meanwhile, babies feel much more constrained when they start moving this way (using all four limbs) because their central nervous system is less flexible.

Another study [7] examined military crawling biomechanics using IMU sensors in 33 healthy adults. The parameters studied were crawl speed, crawl stride time, ipsilateral limb coordination, and contralateral limb coordination. It was found that the group could be divided into two subgroups according to the level of performance: good and low. Good performance is characterized by faster execution and better coordination between the limbs.

In fact, it is a rare occasion that crawling is studied in terms of its biomechanics. From the literature available, we can only know about studies performed based on hands and knees crawl [8,9]. Meanwhile there has been no examination of the biomechanics of military crawling. The difference between these types is significant and quite obvious. In the first case, it is walking on hands and knees [10,11] in real dimensional space. In the second case, it is moving on a horizontal surface. The English term for both cases is crawl, which also refers to the locomotion of swimming in the crawl style—its biomechanics is being studied as well [12].

We consider the locomotion of military crawling based on a specific motor proficiency test which has the following peculiarities: It is a phylogenetic ancient locomotion which has its own automatic behaviors for both lower and upper limbs and trunk. This type of movement is relevant to a certain stage of ontogenesis during the first year of its existence. However, further on in life, it has no particular importance and is mostly not applied. These peculiarities let us consider the locomotion of military crawling as a potentially useful test which makes it possible to detect hidden forms of motor pathology. We assume that the high and low degree of performance, found in [7], even in healthy subjects, demonstrates the high sensitivity of this test and its biomechanical parameters for detecting early forms of motor control disturbances.

Additionally, the process of military crawling on a formal level may be defined as moving mostly in two dimensions (the third vertical dimension is not obvious). This makes the locomotor analysis technically less complicated and may be considered as a 2D one.

The research hypothesis is as follows: We assumed that crawling locomotion, being phylogenetically ancient in humans, remains well automated. This is also true for adulthood and in the case when this type of locomotion is not used in later life in any way (except for the first year, where this is one of the stages of motor development). Symmetry of parameters (left-right) and reciprocity should be relevant to automatic quadripedal locomotion.

In the available literature [7–9], specific questions of the biomechanics of movement during crawling were studied separately. Nevertheless, military crawling as an integral quadripedal process in its kinematic part, has not been studied sufficiently.

The aim of this research is to study the locomotor kinematics of military crawling in accordance with the fixed normal parameters.

## 2. Materials and Methods

### 2.1. Participants

There were 8 healthy participants tested, aged between 15 and 31 (average age 23 years old)—4 men and 4 women without any pre-existing locomotor traumas or dysfunctions. The average height was 178 cm (ranging between 170 and 184 cm) and all subjects were a normal weight (average BMI—22.7, ranging between 21 and 24). In this examination, one of the participants was left handed. All participants had no professional training experience. All the participants had normal physical development. Three participants had no sports training experience. Three others had trained in the past, but by the time the study starting the training break was more than 1 year. Two of the participants trained systematically. All the participants signed informed consent forms (they confirmed their voluntary intentions of participation in the study and consent of wearing registered markers on the body). All the participants had a good state of health.

### 2.2. Military Crawling

A biomechanical analysis of motions used in military crawling was performed. Before the recording started the subject was laid face-down on a special carpet surface with legs joined, arms along the body. Then, the participant had to adopt the initial position: lie down on the floor (the floor has a soft surface), legs straight, arms bent and placed in front of the subject. The task for the person is to crawl 4 m on his/her belly by means of legs and arms at any pace.

Before placing reflective markers, each subject tried to crawl 2–3 times. This was done in order to make sure that the subject would perform exactly what was needed, and not other options, for example, walking on fours. That was a common mistake. However, we deliberately did not allow crawling to be trained for the reason that we were interested in how automatic, symmetrical, and reciprocal the crawling locomotion, which has not been used for many years, would be. We made 2–3 recordings of the crawling process. The recording with the highest quality of registration was selected for subsequent analysis.

*2.3. Equipment*

The kinematics were recorded by the Optitrack optical system ("Motion Capture Systems". OptiTrack, http://www.optitrack.com/index.html, accessed on 3 May 2021) which consists of 12 Flex 13 cameras, 100 Hz each. The cameras are placed at a height of 2.2 m evenly around the perimeter of the room, which was $5 \times 8$ m. The examination was arranged based on the standard procedure: calibration and marker fixation. For the participants there was a set of 13 light-reflecting markers fixed on the following joints (1 marker on each joint): shoulder, elbow, radiocarpal, knee and ankle joints. The remaining 3 markers were fixed on the subject's back: 2 markers on the left and on the right at the level of the sacroiliac joints and 1 on the 10th spinous process of vertebra (Figure 2).

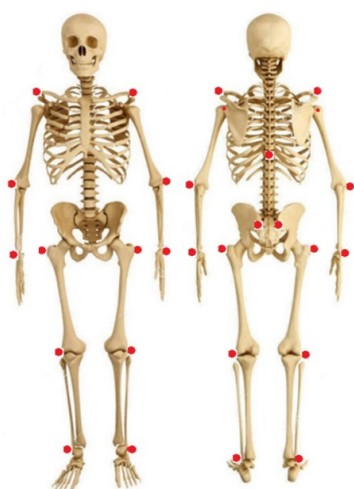

**Figure 2.** Points where markers were fixed: acromion, lateral elbow epicondyle, external place of wrist (watches wrist placement), spina iliaca posterior superior, trochanter major, lateral epicondyle of the knee, lateral malleolus.

Two particular markers are connected with a straight line thus forming the links of the model. The links form joints. Thereby the angle in the joint is determined by three markers (Figure 2).

The crawling cycle is considered to be from the moment the leg is bent in the knee and hip joints until the next knee and hip flexion of the same leg (Figure 3).

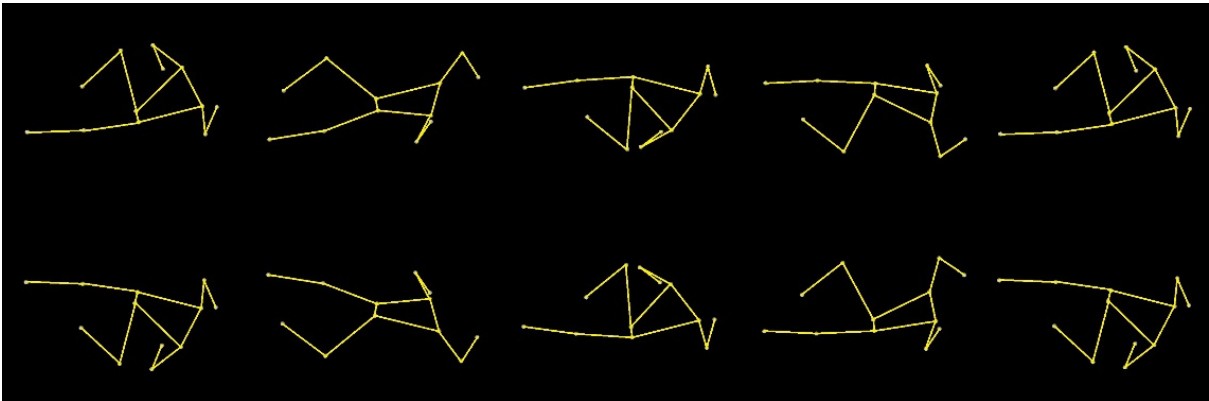

**Figure 3.** Sequential phases of a full cycle of motions for the right side—upper row and left side—lower row.

Since the model is somewhat simplified, the obtained goniograms only partly reflect the joint movements. These movements are studied only in the projection of a horizontal surface. However, to make the description less complicated, we will keep calling them hip, knee, shoulder and elbow joints.

After the records made with the Optitrack system have been processed in the standard way, further processing was performed based on Visual 3D (C-Motion Inc., USA). For the biomechanical analysis of the military crawl task, there has been a special simplified multilink model of a human body developed, which consists of eight links which form eight joints in the horizontal plane.

The beginning and end of each motion cycle on each side (right and left) are determined manually based on the data about the shift of the markers fixed on the lateral malleolus. After that a report containing one-cycle kinematics parameters was generated.

The report includes the following standard biomechanics information: a movement pattern of the marker fixed at the level of the 10th spinous process of vertebra, cycle duration, proportion between the beginning of one cycle and beginning of the next cycle (in percentage), diagrams of angle variance within a movement cycle (meters per second), space-time parameters: movement cycle duration (CD) in seconds, start of a cycle on the opposite side (SOpS) versus the one on the given side in percentage of CD, length of a motion cycle (LMC, meters) and speed of the 10th spinous marker's shift—V (meters per second).

To describe the shoulder girdle—pelvis arrangement, a special value shoulders–pelvis was used. This value is an adjoining angle formed by the lines going through the markers placed on the acromial extremities of the clavicles and markers on the pelvis.

The data set includes the following parameters: range of motion—an amplitude for each joint from minimum to maximum and extremum phase (maximal flexion of a joint)—$T_{max}\%$ (percent of CD).

Statistical processing was performed based on Statistica 10 with the use of variation statistics methods and variance analysis. Evaluation was conducted according to Wilcoxon–Mann–Whitney criteria. These criteria was used because the distribution of parameters in the study was not parametric. This is partly due to the genesis of the values themselves, and partly to the sample size (8 participants).

### 3. Results

The typical motion pattern for 10th spinous process of vertebra marker is shown below (Figure 4). This path looks like a saw-tooth curve. Meanwhile "to the left-to the right" motions are symmetric, and the forward movement is progredient and non-stop. Thereby, the speed does not change significantly.

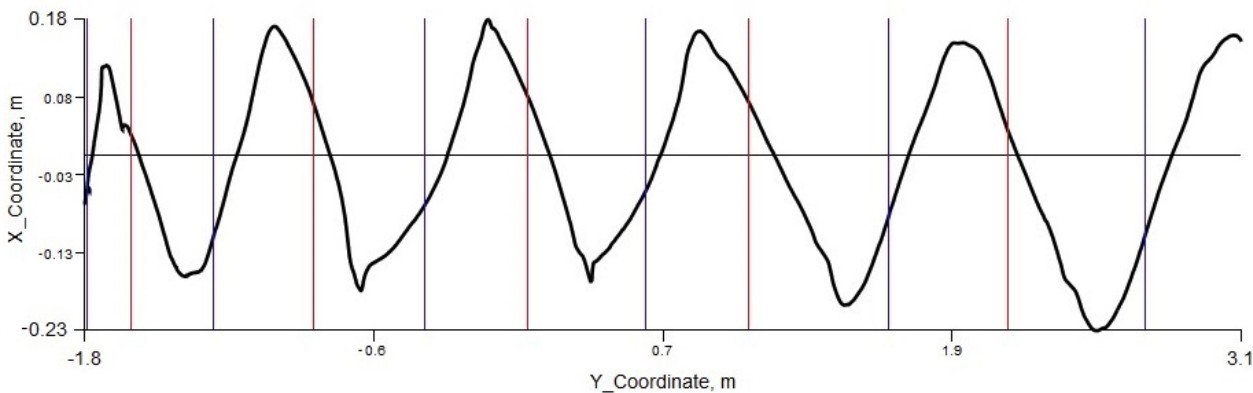

**Figure 4.** Typical motion path of 10th spinous process of vertebra marker. Vertically—motions to the left (up) and to the right (down), scale in meters. Horizontally—motions from the left to the right (scale in meters).

Results of the motion cycle test of space-time parameters are shown in Table 1.

The above findings show that the parameters for the right and left side are relatively symmetrical in terms of both the cycle duration and the covered distance and speed. The shift of motion phases in one cycle versus another fluctuates within 50%, that is a half-period shift. Motion angles test results for healthy participants are shown in Figure 5.

**Table 1.** Space–time parameters of the tested group, *p* > 0.05.

| Parameter | Motion Cycle (CD c.) | Opposite Side Start (SOpS, Percent of CD) | Length of Motion Cycle (LMC, m) | Speed (m/s) |
|---|---|---|---|---|
| Left side | 2.2 ± 0.8 | 47.8 ± 4.9 | 1.2 ± 0.1 | 0.5 ± 0.1 |
| Right side | 2.2 ± 0.8 | 52.3 ± 4.4 | 1.2 ± 0.8 | 0.5 ± 0.1 |

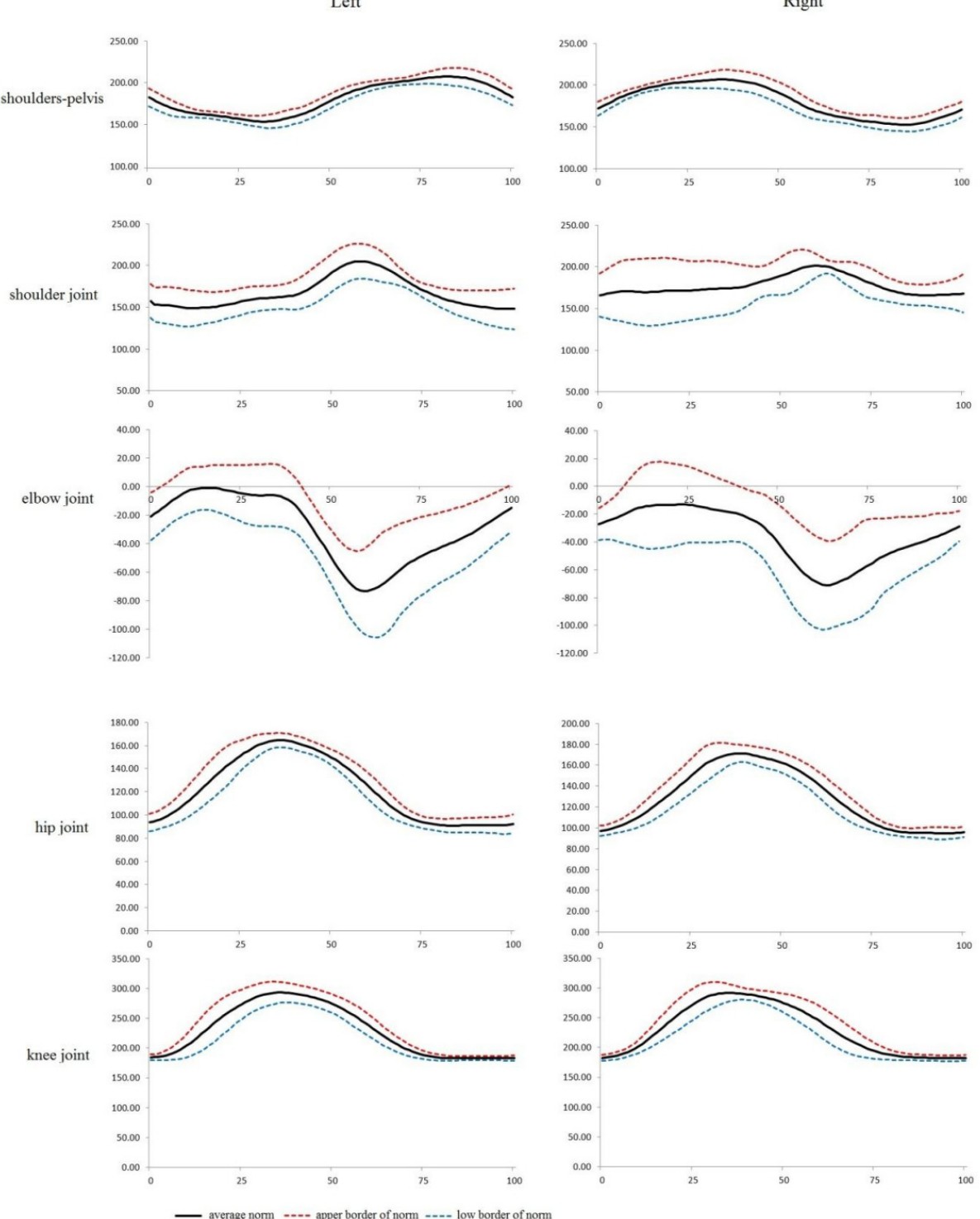

**Figure 5.** Diagrams of activity in joints and body motions on the left and right side in crawling. Horizontally—motion cycle in percentage, vertically—angle in degrees.

The left and right sides are functioning in a relatively symmetric way, but the motions in the right elbow joint have a slightly lower flexion amplitude. Amplitude phase characteristics of motions are given in Table 2.

**Table 2.** Amplitude-phase characteristics of motions for the left and right side. Means and standard deviations are presented. Comparison was executed by Mann–Whitney nonparametric criterium.

| Parameter | Motion Cycle | Max, Grad. | $T_{max}$%, Percent of CD | Mann–Whitney, Z, $p$ |
|---|---|---|---|---|
| Shoulders– pelvis | Left | 56.0 ± 14.4 | 78.5 ± 7.8 | 6.4, <0.0001 |
| | Right | 57.1 ± 14.2 | 31.6 ± 12.0 | |
| Shoulders | Left | 78.4 ± 45.0 | 56.9 ± 3.3 | 0.08, <0.4 |
| | Right | 75.1 ± 37.1 | 51.3 ± 16.9 | |
| Elbows | Left | 89.8 ± 38.3 | 59.4 ± 3.7 | 0.58, <0.56 |
| | Right | 89.3 ± 45.7 | 56.0 ± 16.8 | |
| Pelvis and hips | Left | 79.3 ± 8.3 | 33.6 ± 5.0 | 0.06, <0.95 |
| | Right | 79.0 ± 12.3 | 36.1 ± 5.0 | |
| Knees | Left | 116.9 ± 15.4 | 32.9 ± 7.1 | 0.48, <0.63 |
| | Right | 117.0 ± 7.7 | 35.1 ± 6.1 | |

Here, it is shown (Table 2) that the amplitude and phase of their occurrence for the left and right side are relatively symmetric. The curve shoulders–pelvis has two obvious extremums at 32 and 79 percent of the motion cycle. At these moments the angle between the shoulder-girdle and pelvic-girdle achieves a peak value. However, in one case the angle opens to the left, in the other case it opens to the right. The flexion amplitudes of hip joints and knee joints reach the peak values approximately at the same time, which is 34–36 percent of the motion cycle, while the knee joints' amplitude is 27–28 degrees higher than that in hip joints. The peak amplitudes of shoulder and elbow joints are also almost synchronized and occur at an interval of 51–59 percent of the motion cycle. It is obvious that for them the maximum phase variability is higher, moreover it occurs much later than that of the lower limbs' joints.

## 4. Discussion

An objective, descriptive analysis of military crawling was conducted in the study using the method based on Optitrack motion capture system. In fact, a motion cycle has a different variability because it is arbitrary. If in the case of walking it is evaluated by resonant characteristics of a pendulum represented by lower limbs, in the former case this influence is excluded. The Motion Cycle Start parameter is almost symmetric and fluctuates at about 50 percent of the motion cycle, which is predictable. These are the values characteristic of two-sided cyclic locomotion and walk, particularly in the case of studies [11]. Motion cycle duration is symmetric for both sides too, though the value for average squared difference for the left side is higher. Meanwhile the motion cycle duration is almost similar to a normal cycle length of a step [12]. It sounds quite unexpected, but if the cycle length of a step while walking is an automatic index that normally cannot be controlled on purpose, we may suppose that the same mechanism of control is used in this case as well. However, this phenomenon might be occasional and requires further studies. The speed parameter is also expected to be symmetric, as long as it is measured by the 10th spinous process of vertebra, and normally there must be no significant asymmetry of the left and right sides for this parameter. When analyzing the motion path of the 10th spinous process of vertebra marker, we may find that the norm implies a propulsive progredient motion forward. This is the direction that prevails. Furthermore, there are additional symmetric right–left motions with high reciprocity. The nearest analogue for these additional motions may be found in biomechanics of a normal walk. The center of gravity of the body is also a curve similar to a sinusoid [13]. The

differences are quantitative—the proportion of sideways motions to the forward motion in crawling is close to 1:5, but in walking the step length is much bigger than fluctuations in the bodies' center of gravity for an adult, and the proportion is approximately 1:14. For more information, the speed of the 10th spinous process of vertebra marker was analyzed. For all the tested participants, the diagram looks like a two-phase sinusoid. Meanwhile, the speed varied by 0.2–1.0 meters per second and never reduced to nil, which relates to a full stop. It is noticeable that speed–time curves are in fact similar to those in walking both in terms of amplitude and phase described in this work [13]. Normally, shoulder–girdles and hip–girdles change the angle with respect to each other by 30 degrees, thus forming an angle open to the right or to the left depending on the motion phase. Respectively for the left and right cycle motions will be reverse-phased, because the body represents an integral structure that distributes its motions among the limbs on both sides. Therefore, the goniogram of the angle between the hip–girdles and shoulder–girdles looks like a smooth sinusoid with breakdown by left and right side cycles. The biggest challenge was caused by the phenomenon of high variability of motions in upper and lower limbs' joints during military crawl. To find out the consistency, we have analyzed the variability ratio for goniograms. Variations for each of the goniograms are shown in Figure 6.

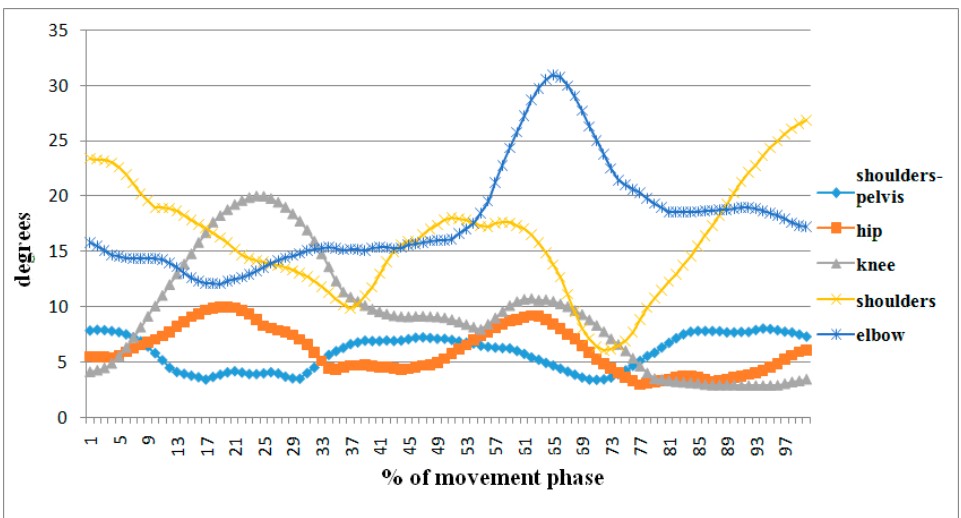

**Figure 6.** Variation values (average absolute deviation) for each goniogram in the format of a motion cycle.

Minimum values of variation were obtained for shoulders–pelvis and hip joints goniograms. The values are higher for knee joints especially in the phase prior to full flexion, that is to say the flexion itself is variable. There is an even higher variation for shoulder joints and the peak one in elbow joints. At the same time the locomotor strategy at the interval of 50–80 percent motion cycle is significantly different (before, during and after the amplitude maximum). For shoulder joints the values vary down at first, then go up; and for elbow joints—vice versa. Furthermore, we estimated average variation values (based on the step cycle). The result confirms the graphic one. Average values come in the same ascending order: shoulders–pelvis—5.9, hips—6.0, knees—9.3, shoulders—16.0, elbows—17.9. A similar calculation was performed for the neighboring joints on the left and right side. The result shows that goniogram variation values of the left side are much lower than those of the right side which is significantly high ($Z = 4.49$, $p < 0.0001$). The only examined left-handed participant had an inverse relationship. More variability was presented in parameters on the left and less on the right. As we suppose, in the phylogenetic ancient locomotor cycle such as crawling the automation of motions has a greater impact on the left side that is less trained. For the more trained right side of the body, voluntary control turns out to be higher. However, this phenomenon requires further study. There are certain physiological pre-conditions for the observed data, namely availability of foot–walk

motions generator [14]. The fact that the activity of this structure influences this locomotion as well (though in a modified way) is proven by a high correlation (Pearson coefficient = 0.98, $p < 0.0001$) of hip and knee joints' performance. The maximum flexion for both is at 36–37 percent motion cycle. In terms of kinematics, it is not even a motion of one and the other joint, but a single action (knee and lower leg moved forward) distributed equally among two joints. Control and coordination processes for upper and lower limbs have some peculiarities as far as upper limbs are concerned. Considering the high variability, the maximum amplitudes for both shoulder and elbow joints are close to 56–59 percent motion cycle. Their variability is much higher than that of lower limbs. Additionally, the control process is organized in such a way that the main motions of an upper limb occur much later than those of a lower limb (on average by 20 percent). The latter is related to the fact that upper limbs play a supportive role, while the leading role is played by lower limbs. First, the main motion is executed by a lower limb, then a supportive motion is performed by an upper limb of the same side. The obtained data demonstrate the presence of a number of quantitative and qualitative symmetries in healthy subjects, as well as stable reciprocity. Prior to this study, we could only assume that this is the case. Thus, the very presence of symmetry of the left and right sides with a high degree of reciprocity and a decrease in variability from the periphery to the center makes it possible to have benchmarks for assessing the performance of this test in patients with CNS lesions.

## 5. Conclusions

The above-mentioned data, though it comes from limited resources, may be used as a guide to estimate results obtained by means of this test for patients with different pathologies of central nervous system and locomotor dysfunctions [15]. Military crawl execution could be an interesting model and a test in clinics for the estimation of motion abilities compared to norm ranges. An approach for collecting data for such norm ranges and a first case is described in this article. The obtained results including space–time data let us characterize military crawling in terms of pelvic and lower-limb motions in a way similar to walking but at a more primitive level. At the same time, the upper limbs' activity is shifted to the second half on a motion cycle—as opposed to walking, when upper limbs' motions are steady and symmetric with reference to the step cycle [15,16]. However, as any reciprocal locomotion, crawling of a healthy adult who does not practice this style of locomotion is characterized by the precise synchronization of movements of the limbs and trunk. Since the data variability descends from periphery to center, an express estimation can use data obtained from a fewer number of markers or even from only one, the 10th spinous process of vertebra marker, as well as by means of slow-response sensors. This is an important conclusion that can assist in the organization of experimental routine in studies based on technology of motion capture systems.

## 6. Limitations of the Study

The capabilities of the method are limited by potentially solvable diagnostic tasks. The study involved a small number of participants. We consider our work as a case study, and the main message of the work is to offer an interesting model of human locomotion, which is not widely covered in the literature. At the same time, the application of a modern objective numeric method for estimation of a special type of human locomotion was applied. In further studies, it is possible to increase the sample of participants and reveal more stable and reliable statistics. It is also possible to compare this type of crawling with other well-known types.

The article is positioned as descriptive, escalates the attention to poorly studied type of human locomotion military crawl and shows the application of new objective methods (motion capture system) for the clarification of hidden parameters of complex motion.

**Author Contributions:** Conceptualization, D.S.; methodology, D.S. and A.A.; software, V.A.; validation, D.S., A.A. and V.A.; formal analysis, D.S., A.A. and V.A.; investigation, D.S., A.A. and V.A.; resources, D.S.; data curation, V.A. and D.S.; writing—original draft preparation, V.A. and D.S.; writing—review and editing, D.S. and V.A.; visualization, D.S. and A.A.; supervision, D.S.; project administration, D.S. All authors have read and agreed to the published version of the manuscript.

**Funding:** This research received no external funding.

**Institutional Review Board Statement:** The study was conducted according to the guidelines of the Declaration of Helsinki. Ethical protocol of the study is №8э/15-17 from 27 October 2017.

**Informed Consent Statement:** Informed consent was obtained from all subjects involved in the study.

**Acknowledgments:** We thank Shipilov A.V. for his assistance during data collection.

**Conflicts of Interest:** The authors declare no conflict of interest.

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
