# Peer review of "Experimental Study of Military Crawl as a Special Type of Human Quadripedal Automatic Locomotion"

_applsci, doi:10.3390/app11167666_

Round 1

Reviewer 1 Report

The authors propose an experimental study on the biomechanics of military crawling, which can give an interesting insight into musculoskeletal dysfunctions analysis.
Experiments are conducted on 8 healthy adult patients.

Besides the topic is poorly covered in literature, in [1] an extended study, with thirty-three participants, on military crawling biomechanic parameters, is done. Indeed, the citation to this work seems to be missing in the manuscript. A comparison of the obtained results with [1] should be also presented, and the scientific contribution should be clarified by highlighting the difference with respect to that.
In order to strenght the contribution, I suggest the authors to consider extending the experimental study and/or results analysis by taking into account different classes of participants (e.g. male/famale, right/left-handed).

Here are some punctual comments on the manuscript:
- Line 16->17: missing reference on "Kylapp system"
- Lien 18->34: a figure would help to understand the difference between different crawling types. Moreover, it would be better to clarify at the end, what would be the definition used troughout the manuscript.
- Line 35->37: missing reference.
- Line 107: "start of a cycle on the opposite side" is not really clear to me. Consider extending the explanation, eventually with an example.
- Line 112: "an amplitude for each joint from minimum to maximum and extremum phase ". Same as previous point.
- Table 2: Clarify whether Left/Right, in the table, refer to the motion (motion on the right side), the Parameter (motion of right shoulder) or both.
- Line 195: the information that one participant is left-handed might be moved in the "Materials and Methods" section
- Line 194->195: it would be interesting to add a comment specific for the left-handed participant data.

[1] "Human crawling performance and technique revealed by inertial measurement units" Rachel V.Vitali, Stephen M.Cain, Steven P.Davidson, Noel C.Perkins.

Author Response

Hello! Thank you for your comments! We have done all the corrections of the material according to your questions and suggestions:

- Line 16->17: missing reference on "Kylapp system"

Reference is inserted
- Lien 18->34: a figure would help to understand the difference between different crawling types. Moreover, it would be better to clarify at the end, what would be the definition used troughout the manuscript.

We added figure.
- Line 35->37: missing reference.

The reference was added.
- Line 107: "start of a cycle on the opposite side" is not really clear to me. Consider extending the explanation, eventually with an example.

We changed the text and explanation.
- Line 112: "an amplitude for each joint from minimum to maximum and extremum phase ". Same as previous point.

Changed.
- Table 2: Clarify whether Left/Right, in the table, refer to the motion (motion on the right side), the Parameter (motion of right shoulder) or both.

The table was restructed.
- Line 195: the information that one participant is left-handed might be moved in the "Materials and Methods" section

Done
- Line 194->195: it would be interesting to add a comment specific for the left-handed participant data.

We done the work as a case study. The problem of studing asymmetry is the next step of the work.

[1] "Human crawling performance and technique revealed by inertial measurement units" Rachel V.Vitali, Stephen M.Cain, Steven P.Davidson, Noel C.Perkins.

Thanks a lot for interesting article, we've read and added the discourse in the text of our article.

Reviewer 2 Report

Experimental study of military crawl as a special type of human quadripedal automatic locomotion

General:

This is an interesting study investigating the military crawl biomechanics locomotion. Although the research question is interesting there are many things inside the manuscript that needs to be changed. Thus, the manuscript needs further processing before it can be consider ready for publication.

Here are my main concerns:

  1. Why this study was conducted? The introduction needs a re-construction from the beginning to the end. Authors have to show inside the intro why military crawl needs further investigation and why this study is important to the readers.
  2. Methods are not clear. I suggest using sub-heading to clarify how this study was conducted. The sample size is small with great diversity. Also, a crucial point here is to present the approval by the local University ethics committee. This is necessary for the further procedure of the manuscript.
  3. There are no statistical indicators in the results (p values and z values from the Mann-Witney analysis). This really limits the understanding of the results. Please provide statistical indicators.
  4. Discussion is confusing because of the results. Authors really need to re-construct the discussion and clearly state the main findings as well as clearly discuss these findings with the scientific literature. Also authors have to add limitations.
  5. What is the take home message for the reader?

Abstract:

Although the abstract is well written, there is a need for clarity. There are no numeric data and statistical values. Also, there is no presentation of the methods used.

Lines 1-5: Delete the references. Add the references inside the introduction.

Lines 7-8: More details for the methods used.

Line 6: …study…

Line 9: ….this locomotion. Please be more specific.

Lines 9-11: Numeric data are missing. Statistical values are missing.

Line 12: patients or participants? As far as I can understand authors had healthy participants. Please, refer to them as participants inside the entire manuscript. 

Line 12: …various theologies….?? I am not familiar with the meaning here. Please rephrase the sentence.

Introduction:

The introduction needs more clarity. Authors have to use more references to establish their research question and why it was important to investigate the military crawl in this population. Also, authors have to connect the paragraphs and the meanings with a better flow inside the manuscript. In general, the introduction is weak.

Line 16-17: Please add a reference here.

Lines 14-34: Authors dedicate 2 paragraphs for the terms. However, authors don’t lead to a term to use in the manuscript. Furthermore, these two paragraphs are confusing and the reader might lose his-her interest. Please, reconsider to shrink the two first paragraphs of the intro. Also, references are missing.

Lines 35-39: Please, add references.

Line 40: Rephrase the first sentence.

Line 44: Rephrase the first sentence.

Line 46: What way is that? Please, be more specific.

Paragraph 6: Delete lines 53-55. Add them in the discussion if needed. Focus on the studies 67,8,9 and present them more details.

Line 56: I suggest to authors to refer military crawl inside the manuscript and skip the word belly.

Paragraph 7: I am not quite sure I am following the way of thinking here. How this paragraph connects with the above analysis? Or even with the research question of the manuscript?

Lines 63-67: Authors reach the aim of the study, but there is no presentation of the importance of the study. What new we might learn from this study? Why it is important to evaluate the biomechanics and kinematics of military crawl?

Material and methods:

Methods are lacking in detail and clarity. More details are needed in order to reproduce the study. I suggest to authors to add sub-paragraphs (Participants, experimental procedures, measurements, statistical analysis). Also, authors need to add intra-correlation coefficients with 95%confident intervals and CV’s% for all variables.

Lines 69-70: Body mass, body height, BMI, training experience, health status, informed concern for adults and under aged participants, local University ethics committee number, participation criteria. All these are missing. This limits the publication of authors study.

Line 71: How many times the participants visited the laboratory? Was there any familiarization testing procedure? What was the order of the measurements?

Line 72: Replace patients.

Line 77: …were…

What was the distance between cameras and participants?

Line 82: Th10 please explain the abbreviation and then use it inside the manuscript.

Line 84: Delete …caption...

Figure 1: Points where markers were fixed.

Figure 2: Please explain what …the same… means. Be more specific.

Line 95: A file “was” generated … Also, what is a C3D file? Please clarify.

Line 102: …was generated.

Lines 103-112: Please provide intra-correlation coefficients with 95%confident intervals and CV’s% for all variables. Also, add line numbers.

Lines 113-115: Why a non-parametric analysis was used? What are the criteria for Mann-Whitney in the present study? What is the significance level?

Results:

Results are incomplete without z scores and Assymp. sig. (p values). Also, is the sample size big enough to proceed with Mann-Witney U test? The paragraph of the results has to be dedicated on the presentation of the results only and not in discussion.  

Figure 3: Explain the abbreviation h10 before using it.

Table1: Add statistical significance.

Line 133: Was that significant? If not, then authors cannot use this result to make conclusions.

Table 2: Add statistical significance.

Line 135: Rephrase the first sentence.

Figure 5: Make sure to delete the green lines.

Discussion:

Discussion is confusing. What was the main finding of the study? Why this finding is important? What this finding mean to readers? Are there any limitations for this study? Compare the current findings with other studies.

Question: Are there any Limitations that come along with this study?

Line 195: The p value here is a result and authors have to present it inside the results paragraph first.  

Lines 202-203: What is the r-Pearson for that correlation?

Lines 217-218: In what patients authors would use the military crawl to evaluate different pathologies of central nervous system and locomotor dysfunctions?

Line 220: Authors had only 8 participants from different ages and from both genders. Is this study enough to produce norm ranges?

Author Response

  1. Why this study was conducted? The introduction needs a re-construction from the beginning to the end. Authors have to show inside the intro why military crawl needs further investigation and why this study is important to the readers. Thank you for the comments, we've rewritten the abstract and the introduction. Main idea of the study was the approbation the method on healthy cohort of participants and starting accumulate norm ranges (of course the number of participants were low and this is a case study).
  2. Methods are not clear. I suggest using sub-heading to clarify how this study was conducted. The sample size is small with great diversity. Also, a crucial point here is to present the approval by the local University ethics committee. This is necessary for the further procedure of the manuscript. Methods now are described wider. We also added ethical protocol in the Institutional Review Board Statement part.
  3. There are no statistical indicators in the results (p values and z values from the Mann-Witney analysis). This really limits the understanding of the results. Please provide statistical indicators. Added.
  4. Discussion is confusing because of the results. Authors really need to re-construct the discussion and clearly state the main findings as well as clearly discuss these findings with the scientific literature. Also authors have to add limitations. We've rewritten the discussion.
  5. What is the take home message for the reader? Our article is about approach of the new method to special motion type rarely described in literature. That is also about starting building norm ranges for futher comparison with clinical cohorts. At the same time we've found several intersting phenomena in dependence of motion indicators on time.

Abstract:

Although the abstract is well written, there is a need for clarity. There are no numeric data and statistical values. Also, there is no presentation of the methods used.

Rewritten.

Lines 1-5: Delete the references. Add the references inside the introduction.

Done.

Lines 7-8: More details for the methods used.

Done.

Line 6: …study…

Done.

Line 9: ….this locomotion. Please be more specific.

Rewritten.

Lines 9-11: Numeric data are missing. Statistical values are missing.

We added the values into the table, but some descriptions here are only descriptive.

Line 12: patients or participants? As far as I can understand authors had healthy participants. Please, refer to them as participants inside the entire manuscript. 

Sorry. This is just because we started to try the method in clinics also. Here we speak only about healthy people - participants.

Line 12: …various theologies….?? I am not familiar with the meaning here. Please rephrase the sentence.

Rewritten.

Introduction:

The introduction needs more clarity. Authors have to use more references to establish their research question and why it was important to investigate the military crawl in this population. Also, authors have to connect the paragraphs and the meanings with a better flow inside the manuscript. In general, the introduction is weak.

Rewritten.

Line 16-17: Please add a reference here.

Done.

Lines 14-34: Authors dedicate 2 paragraphs for the terms. However, authors don’t lead to a term to use in the manuscript. Furthermore, these two paragraphs are confusing and the reader might lose his-her interest. Please, reconsider to shrink the two first paragraphs of the intro. Also, references are missing.

Done.

Lines 35-39: Please, add references.

Done.

Line 40: Rephrase the first sentence.

Done.

Line 44: Rephrase the first sentence.

Line 46: What way is that? Please, be more specific.

Rewritten

Paragraph 6: Delete lines 53-55. Add them in the discussion if needed. Focus on the studies 67,8,9 and present them more details.

Done.

Line 56: I suggest to authors to refer military crawl inside the manuscript and skip the word belly.

Rewritten

Paragraph 7: I am not quite sure I am following the way of thinking here. How this paragraph connects with the above analysis? Or even with the research question of the manuscript?

Rewritten

Lines 63-67: Authors reach the aim of the study, but there is no presentation of the importance of the study. What new we might learn from this study? Why it is important to evaluate the biomechanics and kinematics of military crawl?

Thanks again. Yes, this is just case and descriptive study. We fulfill first step in building valid methodology for recording norm ranges for healthy and futher for clinical cohorts of people. Some interesting findings could make sence also and help in futher strategy of development.

Material and methods:

Methods are lacking in detail and clarity. More details are needed in order to reproduce the study. I suggest to authors to add sub-paragraphs (Participants, experimental procedures, measurements, statistical analysis). Also, authors need to add intra-correlation coefficients with 95%confident intervals and CV’s% for all variables.

Done.

Lines 69-70: Body mass, body height, BMI, training experience, health status, informed concern for adults and under aged participants, local University ethics committee number, participation criteria. All these are missing. This limits the publication of authors study.

We added information in the text.

Line 71: How many times the participants visited the laboratory? Was there any familiarization testing procedure? What was the order of the measurements?

We added the description in the text.

Line 72: Replace patients.

Done. Sorry.

Line 77: …were…

Done

What was the distance between cameras and participants?

The cameras are placed at a height of 2.2 m evenly around the perimeter of the room 5x8 m. Participants moved in the central diameter of the circle. Information was added in the text.

Line 82: Th10 please explain the abbreviation and then use it inside the manuscript.

Done.

Line 84: Delete …caption...

Done.

Figure 1: Points where markers were fixed.

Done.

Figure 2: Please explain what …the same… means. Be more specific.

Done.

Line 95: A file “was” generated … Also, what is a C3D file? Please clarify.

This is standard file of Optitrack system. We rephrased the text here.

Line 102: …was generated.

Done.

Lines 103-112: Please provide intra-correlation coefficients with 95%confident intervals and CV’s% for all variables. Also, add line numbers.

This information is provided in tables. We also changed the form of them.

Lines 113-115: Why a non-parametric analysis was used? What are the criteria for Mann-Whitney in the present study? What is the significance level?

Evaluating was done according to Wilcoxon-Mann-Whitney criteria. This criteria was used because the distribution of parameters in the study was not parametric. This is partly due to the genesis of the values themselves, partly to the sample size (8 participants).

Results:

Results are incomplete without z scores and Assymp. sig. (p values). Also, is the sample size big enough to proceed with Mann-Witney U test? The paragraph of the results has to be dedicated on the presentation of the results only and not in discussion.  

All information was added.

Figure 3: Explain the abbreviation h10 before using it.

We've changed the abbreviation. This is 10-th spinous process of vertebra marker.

Table1: Add statistical significance.

This is just descriptive table.

Line 133: Was that significant? If not, then authors cannot use this result to make conclusions.

Table 2: Add statistical significance.

Done.

Line 135: Rephrase the first sentence.

Done.

Figure 5: Make sure to delete the green lines.

Done.

Discussion:

Discussion is confusing. What was the main finding of the study? Why this finding is important? What this finding mean to readers? Are there any limitations for this study? Compare the current findings with other studies.

Thanks again. Yes, this is just case and descriptive study. We fulfill first step in building valid methodology for recording norm ranges for healthy and futher for clinical cohorts of people. Some interesting findings could make sence also and help in futher strategy of development.

Question: Are there any Limitations that come along with this study?

We've added the limitations part in the text.

Line 195: The p value here is a result and authors have to present it inside the results paragraph first.  

Lines 202-203: What is the r-Pearson for that correlation?

The numeric data was added.

Lines 217-218: In what patients authors would use the military crawl to evaluate different pathologies of central nervous system and locomotor dysfunctions?

This work done on healthy participants (sorry, in one place we missed the word), but we also contacted with clinics and offered this method in neurorehabilitation and stroke rehabilitation.

Line 220: Authors had only 8 participants from different ages and from both genders. Is this study enough to produce norm ranges?

Thank you for this comment. Of course the number is really not enough for produce norm ranges. We consider our work as a case study and building methodics for future constructing base for norm. That is the next steps.

Reviewer 3 Report

The paper presents an interesting approach of study for biomechanics of military crawl.

The paper is interesting and well structured.

Hovewer some point should be addressed to improve the soundness of the paper.

More biomechanics tecniques should be mentioned to underline why the once proposed has been used.

You can refere to these paper and mention them.

Chaparro-Rico, B.D.M., Cafolla, D., Castillo-Castaneda, E., Ceccarelli, M. Design of arm exercises for rehabilitation assistance (2020) Journal of Engineering Research (Kuwait), 8 (3), pp. 203-218. DOI: 10.36909/JER.V8I3.6523 Chaparro-Rico, B.D.M., Cafolla, D. Test-retest, inter-rater and intra-rater reliability for spatiotemporal gait parameters using SANE (an eaSy gAit aNalysis systEm) as measuring instrument (2020) Applied Sciences (Switzerland), 10 (17), art. no. 5781, . https://www.scopus.com/inward/record.uri?eid=2-s2.0-85090195968&doi=10.3390%2fAPP10175781&partnerID=40&md5=d5f2beb6cf9f6390ba357888a18f2945 DOI: 10.3390/APP10175781

Figure 2 is too small

Methodology have to be better detailed

Ethics should be added with document number and ethical approvment since subject are involved with wearable markers.

Author Response

More biomechanics tecniques should be mentioned to underline why the once proposed has been used.

Thank you for the comment. We added more information about the method and focused on the point, why we used such type of locomotion and such registration method.

You can refere to these paper and mention them.

 Thank you very much! We read the papers and made references to them by now (and small discussion about their study also).

Chaparro-Rico, B.D.M., Cafolla, D., Castillo-Castaneda, E., Ceccarelli, M. Design of arm exercises for rehabilitation assistance (2020) Journal of Engineering Research (Kuwait), 8 (3), pp. 203-218. DOI: 10.36909/JER.V8I3.6523 Chaparro-Rico, B.D.M., Cafolla, D. Test-retest, inter-rater and intra-rater reliability for spatiotemporal gait parameters using SANE (an eaSy gAit aNalysis systEm) as measuring instrument (2020) Applied Sciences (Switzerland), 10 (17), art. no. 5781, . https://www.scopus.com/inward/record.uri?eid=2-s2.0-85090195968&doi=10.3390%2fAPP10175781&partnerID=40&md5=d5f2beb6cf9f6390ba357888a18f2945 DOI: 10.3390/APP10175781

Figure 2 is too small

The figure was rebuilded.

Methodology have to be better detailed

Added.

Ethics should be added with document number and ethical approvment since subject are involved with wearable markers.

Added in the Institutional Review Board Statement part.

Round 2

Reviewer 2 Report

The abstract should be a total of about 200 words maximum. Please refer to authors instructions and use headings as well. 

Lines 33-35: How the results of the study will support this statement? Authors need to add a relevant text inside the introduction and discussion.

The introduction needs a better flow especially in the last 3 paragraphs. Please add a hypothesis after the aim of the study.

Change the heading "The Task" to "The military crawling".

Was the familiarization session enough for participants to execute in the proper way the military crawl? Normanly, a familiarization session or period may need weeks or a day. Please clarify.

Still the sub-heading of statistics and the p value are missing.

Table 1: Is this m/c meters per cycle?

Table 2: very good.

Pleased with the limitations section.

Author Response

1) The abstract should be a total of about 200 words maximum. Please refer to authors instructions and use headings as well. 

We have reduced the abstract.

2) Lines 33-35: How the results of the study will support this statement? Authors need to add a relevant text inside the introduction and discussion.

We have done it.

3) The introduction needs a better flow especially in the last 3 paragraphs. Please add a hypothesis after the aim of the study.

Added.

4) Change the heading "The Task" to "The military crawling".

Done.

5) Was the familiarization session enough for participants to execute in the proper way the military crawl? Normanly, a familiarization session or period may need weeks or a day. Please clarify.

We've added the description about it in the text.

6) Still the sub-heading of statistics and the p value are missing.

Added in one more point.

7) Table 1: Is this m/c meters per cycle?

That are meters per second. Sorry, changed.

Please, find also our cover letter in the attachment.
